# Different Response Behavior to Therapeutic Approaches in Homozygotic Wilson’s Disease Twins with Clinical Phenotypic Variability: Case Report and Literature Review

**DOI:** 10.3390/genes13071217

**Published:** 2022-07-07

**Authors:** Sara Samadzadeh, Theodor Kruschel, Max Novak, Michael Kallenbach, Harald Hefter

**Affiliations:** 1Department of Neurology, University of Düsseldorf, Moorenstrasse 5, 40225 Düsseldorf, Germany; sara.samadzadeh@yahoo.com (S.S.); theo.kruschel@gmx.de (T.K.); max.novak@uni-duesseldorf.de (M.N.); 2Department of Gastroenterology, University of Düsseldorf, Moorenstrasse 5, 40225 Düsseldorf, Germany; michael.kallenbach@med.uni-duesseldorf.de

**Keywords:** monozygotic twins, epigenetic mechanism, genotype-phenotype discordance

## Abstract

Background: Wilson’s disease (WD) is an autosomal-recessive disorder of copper deposition caused by pathogenic variants in the copper-transporting ATP7B gene. There is not a clear correlation between genotype and phenotype in WD regarding symptom manifestations. This is supported by the presentation of genetically identical WD twins with phenotypic discordance and different response behavior to WD-specific therapy. Case Presentation: One of the female homozygous twins (age: 26 yrs) developed writing, speaking, swallowing and walking deficits which led to in-patient examination without conclusive results but recommended genetic testing. Both sisters were tested and were heterozygous for the C.2304dupC;p(Met769Hisf*26) and the C.3207C>A;p(His1069Gln) mutation. Self-medication of the affected sibling with 450 mg D-penicillamine (DPA) did not prevent further deterioration. She developed a juvenile parkinsonian syndrome and became wheelchair-bound and anarthric. A percutaneous endoscopic gastrostomy was applied. Her asymptomatic sister helped her with her daily life. Despite the immediate increase of the DPA dose (up to 1800 mg within 3 weeks) in the severely affected patient and the initiation of DPA therapy (up to 600 mg within 2 weeks) in the asymptomatic patient after the first visit in our institution, liver function tests further deteriorated in both patients. After 2 months, the parkinsonian patient started to improve and walk again, but experienced several falls, broke her right shoulder and underwent two necessary surgical interventions. With further consequent copper elimination therapy, liver dysfunction improved in both patients, without need for orthotopic liver transplantation (LTX) in the severely affected patient. Her excellent recovery of liver and brain dysfunction was only transiently interrupted by the development of a nephrotic syndrome which disappeared after switching to Cuprior^®^. Unfortunately, she died from fulminant pneumonia. Conclusion: Despite identical genetic disposition, WD symptom presentations may develop differently in monozygotic twins, and they may need to be placed on a very different therapeutical regimen. The underlying gene-environment interaction is unclear so far.

## 1. Introduction

Wilson´s disease (WD) is an autosomal recessively inherited disorder of copper metabolism caused by a defect of the large ATP7B gene localized on chromosome 13 [1,2,3] which contains 20 introns and 21 exons. More than 1000 mutations have been described in ATP7B. [4,5] Those mutations in ATP7B and inactivation of the ATP7B transporter in hepatocytes result in the failure of the biliary excretion of copper, which leads to disturbed copper homeostasis [6,7]. ATP7B is also responsible for transporting copper for the synthesis of functional ceruloplasmin [8,9]. Therefore WD patients suffer from decreased serum levels of ceruloplasmin and total serum copper and increased levels of toxic non-ceruloplasmin-bound copper [10].

Hepatic and systemic overload of toxic copper is the major cause of tissue pathology and clinical symptoms in many different organs including the central nervous system (CNS) in the later stages of WD-symptom presentation [11,12,13]. High levels of copper enter into the brain [14,15] and affect the structure and function predominantly of the grey matter [16,17], especially in the basal ganglia, the cortex and the brain stem nuclei. Typical clinical neurological manifestations result from the involvement of these different brain structures [11,18,19,20].

The challenging task of finding a correlation between the ATP7B genotype and the WD phenotype has been attempted in different studies without much success [21,22,23,24]. An epigenetic mechanism contributes to this genotype-phenotype discordance and can potentially explain different spectrums of symptoms. The weight of different triggering factors causing epigenetic modifications is often unknown. However, it is important to identify them to complement the therapeutic approach, prevent disability and improve patients’ quality of life [25].

Several case reports of homozygous WD twins with different disease phenotypes and also studies in animal models indicate and support the involvement of epigenetic changes in the pathogenesis of WD and its phenotypic presentation, suggesting mostly that environmental or nutritional factors may affect WD [26,27,28,29,30].

Here, we describe another pair of monozygotic WD twins with different clinical presentations, but in contrast to previous reports on monozygotic WD-twins, we will focus more on the difficulties of treatment and management and also demonstrate the differences of disease course and therapy response.

## 2. Case Presentations

### 2.1. History before First Visit to Our Institution

The history of these monozygotic sisters before their first visit in our institution is summarized in Table 1. Until their marriage at the age of 22 there was no hint for the development of a neurological disorder, either in sister 1 or in sister 2.

After her divorce, sister 1 progressively developed neuropsychiatric symptoms which were initially interpreted to be of psychogenic origin. Then, hemochromatosis was suspected, and an inpatient evaluation was performed over three months. Finally, during a further neurological inpatient evaluation, WD was diagnosed because of the presence of Kayser-Fleischer rings. Treatment with 300 mg d-penicillamine (DPA) was initiated and the patient was dismissed with the recommendation of genetic testing, which confirmed WD in both sisters.

### 2.2. History after First Visit to Our Institution

With a significant time lag of about 4 months after the diagnosis of WD in sister 1, the two patients finally presented themselves in our institution on their own initiation.

### 2.3. A History of Sister 1, the Severely Affected Patient

Sister 1 had increased the DPA dose to 450 mg. Nevertheless, she had experienced a rapid deterioration during the previous 4 months. At the first presentation (Figure 1), sister 1 was almost anarthric, had severe difficulties in swallowing, which were compensated by PEG application, had a severe juvenile parkinsonian syndrome with considerable slowness of movement and a severe stance and gait ataxia, so that she was unable to walk alone and had to rely on an accompanying person or a wheelchair (Figure 1). In addition, there was vertical gaze palsy and hypersalivation. Sensory abnormalities or pyramidal tract signs could not be found. Writing was not possible. There was only a mild anxiety disorder, probably on the basis of helplessness, with no evidence of an intellectual deficit.

Due to symptom severity, the daily dose was increased over 3 weeks from 450 mg/day to 3 times 600 mg DPA/day. Initially, the patient responded well to the DPA therapy; the liver values improved significantly, and the copper excretion was initially very high and then decreased as expected. The neurological symptoms started to improve. Then, after about 6 months, the patient developed weight gain and dyspnea and a further deterioration in her general condition. The improvement in the neurological symptoms stagnated. In view of the multiplication of protein excretion, a nephrotic syndrome was diagnosed as a drug side effect due to DPA (Figure 2C (left side; triangles)). The patient was switched from 3 × 2 tablets DPA 300 mg to 3 × 2 tablets Cuprior^®^ 150 mg within 6 days by exchanging 1 tablet DPA to 1 tablet Cuprior^®^. After this switch, the protein loss and the general condition, especially the dyspnea and the neurological symptoms, further improved (Figure 2A,C (left side)).

With the increasing mobility of the patient—the patient was finally able to walk a few steps without help—unfortunately, she experienced a fall with a broken shoulder on the right and two necessary operations as a result, which further delayed the rapid neurological improvement. After this, she was able to go on a long-distance trip for two months to visit her grandparents.

Upon her return, the patient developed a fever and cough and was admitted to a peripheral hospital. This happened at the time of the third wave of the COVID-19 pandemic. Allegedly, the patient tested negative, but we have no documentation of the COVID-19 test result. An autopsy was not carried out due to the natural death of the patient. In the differential diagnosis, bronchopneumonia should be considered as the cause of death in asphyxia despite percutaneous endoscopic gastrostomy (PEG) application.

### 2.4. B History of Sister 2, the Mildly Affected Patient

Clinically and neurologically, the second sister was asymptomatic at the first presentation (Figure 1, standing person; Figure 2A, right side). She had a body weight of less than 50 kg. DPA therapy was initiated and increased over 2 weeks up to 600 mg.

Although this patient was clinically asymptomatic, she already had abnormal liver values, which worsened even further in the initial phase of the dose increase. After a few weeks, however, there was a clear improvement in the liver values (Figure 2B, right side) To prepare for pregnancy, the DPA dose was increased up to 1500 mg. After a positive pregnancy test, the DPA dose was reduced to 300 mg. During the ongoing chelating therapy and pregnancy, she remained asymptomatic (Figure 2A, right side).

### 2.5. Clinical Scores, Liver Enzymes, Urinary Copper and Protein Excretion in Sister 1 and 2 Demonstrating the Development of a Nephrotic Syndrome in Sister 1

At each visit both sisters underwent a detailed clinical neurological investigation. Seven motor symptoms (dystonia, dysarthria, bradykinesia, tremor, gait disturbance, oculomotor deficits, ataxia of extremities) as well as three non-motor symptoms (reflex abnormalities, sensory symptoms, neuropsychological and psychiatric symptoms) were scored as to whether these symptoms were absent (0) or only mildly (1), moderately (2) or severely (3) present. The motor scores were summed up to yield a Wilson motor score (MotS: 0–21), the three non-motor items were summed up to give a non-motor score (N-MotS: 0–9) and the sum of MotS and N-MotS yielded the total score (TSC: 0–30). These scores have been used in our department to monitor therapy in WD for about 30 years now [31] and are similar to the score used in the Italian Monotematica AISF 2000 OLT study group [32]. WD patients are scored to be mildly affected as long as TSC <3, are scored moderately affected with TSC between 3 and 6 and are scored to be severely affected with TSC >6. In Figure 2A, the temporal development of MotS (circles), N-MotS (triangles) and TSC (squares) are presented for sister 1 on the left and sister 2 on the right side.

Out of a long list of laboratory parameters being determined during our routine therapy monitoring of WD the following parameters were selected to demonstrate efficacy in copper chelating therapy and the difference in response to therapy of the twins: the liver enzymes ALT (circles), GGT (triangles) and CHE (squares) (Figure 2B) and the 24 h urinary excretion of copper (circles) and proteins (triangles) (Figure 2C).

### 2.6. cMRI-, OCT-, ARFI-, US-Investigation and MELD-Scores in Sister 1 and 2

Both sisters underwent further investigations which confirmed severe impairment of brain and liver in sister 1 and normal brain imaging and mild liver involvement in sister 2 (Figure 3).

The optical coherence tomography (OCT) examination revealed a normal size optic disc with almost normal retinal nerve fiber layer (RNFL) in both eyes of the two patients. The specific pattern of change in WD with the reduction in diameter of RNFL, GCIP and INL could not be seen clearly in segmented layers. There were slight differences between the OCT parameters of the two patients, but they were still in the normal range in comparison with normal control (Table 2).

## 3. Discussion. 1. Previous Reports on Homozygous WD Twins in the Literature

Several case reports on homozygous WD twins with a different clinical phenotype have been presented [26,27,28,29,30]. Senzolo et al. [26] described two homozygotic twins, both with liver cirrhosis due to WD, one of them with severe neuropsychiatric involvement. Both underwent liver transplantation but subsequently had a very different outcome. Członkowska et al. [27] examined two pairs of monozygotic twins discordant for the WD phenotype and suggested that the phenotypic characteristics were attributable to epigenetic/environmental factors. Details on medication were not presented. Kegley et al. [28] reported a case of one monozygotic twin presenting with acute liver failure requiring emergent liver transplantation, while the other twin presented with mild liver disease. Nan Cheng et al. [30] also studied the clinical and genetic characteristics of 5 pairs of WD twins and emphasized the possibility of phenotypical discordance [30] but did not describe differences in phenotype and treatment in detail (an English translation of this paper was available). Therefore, these cases were not included in Table 3.

### 3.1. Lack of Pheno/Genotype Correlation at Onset of Symptoms

Clinical symptoms in WD result from tissue damage due to copper intoxication. Copper intoxication is dependent on duration of exposure to and the amount of toxic copper which is penetrated into the tissue of an organ. However, damage of an organ in WD is not only dependent on the transport of copper to the organ, but also on the ability of the organ to regenerate. The liver, e.g., is exposed to high amounts of copper in WD long before other organs are involved, but it has a high capability for regeneration. On the other hand, the central nervous system is well protected against copper for a long time by the blood brain barrier. Nevertheless, it is highly vulnerable to heavy metal intoxication [37].

Differences in time to diagnosis and differences in nutrition and lifestyle all add to the variability of the phenotype. More than 1000 different mutations of the ATP7B gene cause a high genotype variation [4,5]. Therefore, it is no surprise that there is no clear-cut geno/phenotype correlation in WD.

In our WD twins no difference in their disease development is obvious until their marriages. After marriage, both sisters lived at different places and sister 2 became pregnant soon after her marriage, which (at least theoretically) may have been an unusual treatment of beginning WD. We therefore think that differences in lifestyle and pregnancy may have led to a difference in the clinical manifestation of WD. That WD might also have become manifest in sister 2 can be seen from the elevated liver enzyme levels which further deteriorated even when WD-specific treatment was initiated (Figure 2B, left side).

### 3.2. Lack of Pheno/Genotype Correlation after Onset of Therapy

When WD-specific therapy is initiated, the situation becomes even more complex. Patients are treated with different drugs and different doses of a specific drug. Some symptoms respond better to therapy in WD than others [38,39]. Therefore, it is important to monitor the treatment effect carefully.

To achieve improvement of neurological symptoms, higher doses of DPA are probably necessary than for the improvement of hepatic symptoms. We have observed a secondary worsening of glucose metabolism of the CNS dependent on the duration of treatment with less than 1000 mg DPA [40]. We have therefore recommended a dose of at least 1200 mg DPA is used in WD patients with a moderate to severe neurological manifestation [40]. This is a possible explanation for the different response to 1000 mg/day in two sisters with the same genotype observed by Sapuppo et al. [41]. Under this therapy the sister who suffered from neurological symptoms further deteriorated, whereas the other sister with a hepatic manifestation improved.

Sister 1 was treated with much higher DPA doses than sister 2. In principle, high doses of DPA can lead to further worsening of neurological symptoms. However, we do not think this happened in sister 1, since she started to improve before the development of the nephrotic syndrome. Anyway, differences in the response to drugs can further contribute to a different long-term outcome in WD.

Unfortunately, details of therapy have only been reported for a few cases of homozygotic WD twins with a phenotype discordance (see Table 3). This makes an explanation of the different disease courses in WD twins with identical genotypes, but different phenotypes, nearly impossible.

### 3.3. Possible Influence of Epigenetic Factors

The potential role of epigenetic mechanisms has been emphasized [25] and explored in animal models of WD [29]. In WD rats, a high-calorie diet aggravates mitochondrial dysfunction and triggers severe liver damage [29].

At the interface between the regulation of gene expression and the environment is methionine metabolism, a metabolic pathway that has regulatory effects on DNA methylation. The enzyme S-adenosylhomocysteine (SAH) hydrolase (SAHH; also known as AHCY) has a crucial role in methionine metabolism as it is responsible for metabolizing SAH to homocysteine. If the expression or activity of SAHH are decreased, the level of SAH, which acts as an inhibitor of DNA methylation reactions, will increase. Importantly, SAHH enzyme activity and gene (AHCY) transcript levels are decreased in the presence of hepatic copper accumulation with consequent downstream changes in methionine metabolism parameters [21,22].

Notably, the toxic milk mouse from the Jackson Laboratory (tx-j mouse), which has a spontaneous point mutation affecting the second transmembrane region of the copper transporter, showed dysregulation of methionine metabolism and global DNA hypomethylation in hepatocytes [23,24], with possible downstream effects on the regulation of genes involved in the development of liver damage. In addition, during embryonic development, the liver (a site of major methylation rearrangements) presented major changes in gene transcript levels related to cell cycle and replication in tx- j mice compared with control animals [24]. The provision of supplemental methyl donor choline to pregnant mice was able to bring gene expression in embryonic mice to the same levels as control animals, indicating that fetal livers are susceptible to nutritional factors with potential lifelong consequences for disease phenotype and progression [24].

### 3.4. Possible Influence of Modifier Genes

So far, WD is thought of as a monogenic disease. However, additional involvement of modifier genes cannot be excluded, as long as whole genome analysis has not been performed in larger cohorts of WD patients. In a recent analysis of clinical findings in a large cohort of long-term-treated WD patients, a broad spectrum of comorbidities has been described with the possibility of mutual interaction with WD [42]. This is summarized in a graphical scheme.

## 4. Conclusions

There is no clear-cut geno/phenotyp correlation in WD, either before or after WD-specific therapy. This is supported by the observation that homozygotic WD twins may present with totally discordant phenotypes and a totally different response behavior to therapy as in the cases presented above. The reasons are poorly understood and range from differences in lifestyle and compliance to a complex metabolic influence on gene regulation.

## Figures and Tables

**Figure 1 genes-13-01217-f001:**
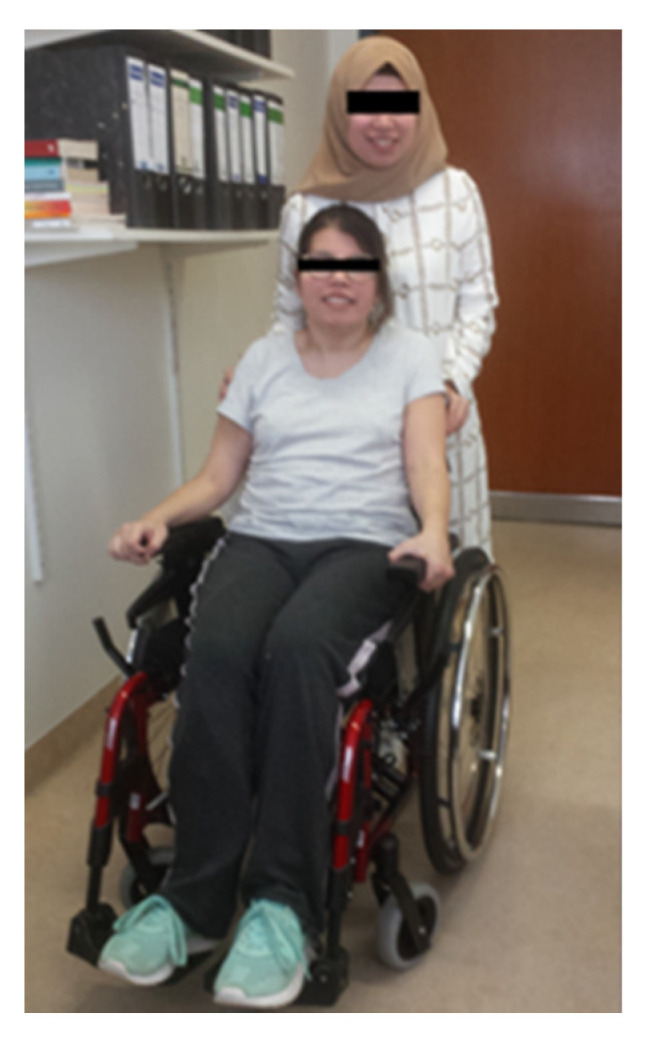
These homozygotic twins are clinically affected in completely different ways: one sister is wheelchair-bound, the other sister is asymptomatic. Both patients responded to DPA therapy, however, the severely affected patient developed a nephrotic syndrome during DPA therapy.

**Figure 2 genes-13-01217-f002:**
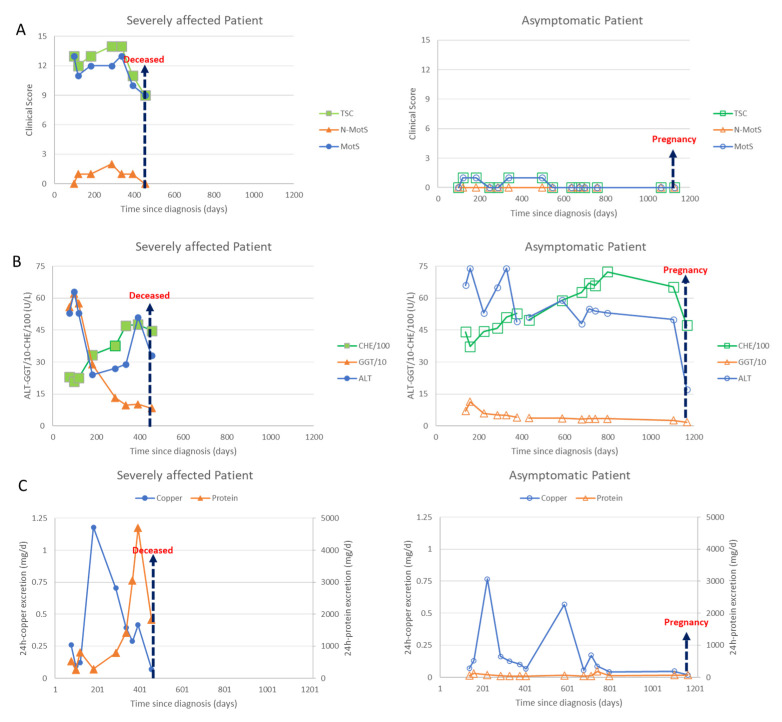
Data of the severely affected patient are presented on the left side (full symbols) and those of the asymptomatic patient on the right side (open symbols). Arrows indicate death of the severely affected twin or second pregnancy of the asymptomatic twin. (**A**) In Figure 2A clinical scores are presented. The motor score (MotS; circles), the non-motor score (N-MotS; triangles) and the total score (TS; squares) clearly improve after less than 10 months of adequate treatment (for details see case report). The scores of the asymptomatic patient were less than 2 all the time. (**B**) In Figure 2B liver enzymes are presented. The values of the alanin-aminotransferase (ALT; circles) are presented without scaling, the values of the gamma-glutanyl-transferase (GGT; triangles) were divided by 10 and the values of the pseudocholinesterase (CHE) were divided by 100. About 100 days after onset of treatment all liver enzymes started to improve in both patients. (**C**) In Figure 2C the daily excretion of copper (circles) and protein (triangles) in the 24 h urine are presented. Adequate copper chelating treatment highly increased the daily copper excretion in both patients. The 24 h copper excretion slowly declined with duration of treatment. After 7 months of sufficient treatment with DPA a sudden increase of the protein excretion was observed in sister 1 due to a nephrotic syndrome which could be stopped by switching medication from DPA to Cuprior^®^. Despite the development of a nephrotic syndrome, the neurological symptoms improved in the severely affected twin (compare Figure 2A,C (left side)). After the death of her sister, the asymptomatic patient (right side) interrupted intake of medication, which led to a second transient peak in the copper excretion.

**Figure 3 genes-13-01217-f003:**
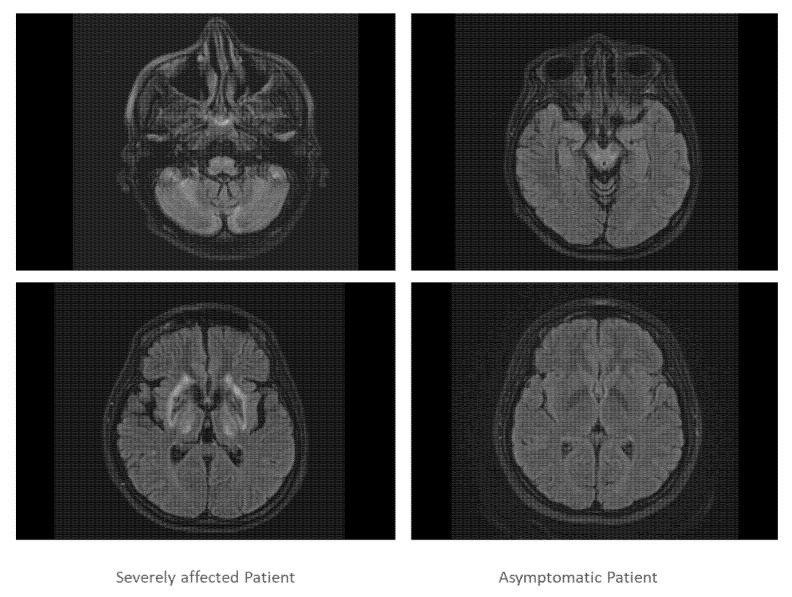
In Figure 3, two slices of the cranial MRI scan of the severely affected twin (**left** side) and the asymptomatic twin are presented (**right** side). The MRT scan of the severely affected twin demonstrated that cortex, basal ganglia, cerebellum and pontine nuclei were heavily involved in the disease process, whereas the MRI scan of the asymptomatic patient was normal.

**Table 1 genes-13-01217-t001:** History of sister 1 and 2 before the first visit in our institution.

	Sister 1 (Severely Affected Twin)	Sister 2 (Asymptomatic Twin)
place of residence	until her marriage at the age of 22 in her parent´s house	until her marriage at the age of 22 in her parent´s house
childhood	normal milestones	normal milestones
school	normal school, vocational training	normal school, vocational training
marriage	after marriage development of health problems, divorce after 2 yrs, return to her parent´s house	lived with her husband at another place, pregnancy soon after marriage, delivery of a healthy son
disease development	development of tremor, difficulties in writing, speaking and swallowing as well as a gait disorder	remained healthy
general inpatient evaluation	hemochromatosis suspected, but liver biopsy was not conclusive	n.a.
neurological inpatient evaluation	normal lumbar puncture Kayser-Fleischer rings were detected	n.a.
genetic testing	C.2304dupC;p(Met769Hisf*26) & C.3207C>A;p(His1069Gln) mut.	C.2304dupC;p(Met769Hisf*26) & C.3207C>A;p(His1069Gln) mutation
treatment	d-penicillamine (DPA): 300 mg	no treatment

**Table 2 genes-13-01217-t002:** Different clinical investigations in sister 1 and 2.

Investigation	Sister 1 (Severely Affected Twin)	Sister 2 (Asymptomatic twin)
cMRI	wide-spread impairment (comp. Figure 3 left side)	normal (Figure 3 right side)
OCT	normal	normal
ARFI	2.32 m/s (LFS4: cut-off >= 1.76) definite liver cirrhosis [33]	1.25 m/s (<LFS2: cut-off < 1.27) normal finding [33]
US	many echo-reduced knots, moderate to severe liver cirrhosis, HCC not excluded	liver slightly enlarged, several regenerative knots, beginning liver involvement due to WD
MELD-score	11	6

cMRI = cranial magnetic resonance imaging; OCT = optical coherence tomography; ARFI = acoustic radiation force impulse elastography [34,35]; LFS = Ludwig fibrosis score (0–4) [33,35,36]; US = abdominal ultrasound investigation; HCC = hepatocellular carcinoma; WD = Wilson’s disease.

**Table 3 genes-13-01217-t003:** Further homozygotic twins described in the literature.

Investigation	Severely Affected Twin	Less Affected Twin
Publication 1: Senzolo et al. [26]		
genetic testing	heterozygous for two different mutations (A1183G/R1319X)	heterozygous for two different mutations (A1183G/R1319X) of ATP7B
clinical manifestation, physical examination	bleeding of esophageal varices the extrapyramidal symptoms refractory ascites mild encephalopathy tremor—slurred speech ataxic gait neuropsychiatric disease	ascites portosystemic encephalopathy mild dysarthria neuropsychiatric disorder associated with drug abuse
slitlamp (KF-ring)	complete KF-ring	incomplete KF-ring
EEG	bitemporal theta activity	mild bitemporal theta activity
CT	basal ganglia hypodensity	no anatomical lesions
EMG	sensory and motor alteration of the left median nerve	-
SPECT	hypoperfusion of the basal nuclei and thalamus	mild tracer defect in both occipital lobes, otherwise normal
treatment	zinc sulphate (440 mg t.i.d.) poor compliance	zinc sulphate (440 mg t.i.d.)
liver transplantation	died after 2 months	successful
Child-Pugh score	C10	C11
Publication 2: Członkowska et al. [27]		
Twin pair 1		
genetic testing	heterozygous for c.3207C>A (p.H1069Q) and c.1211_1212insA (p.N404Kfs)	heterozygous for c.3207C>A (p.H1069Q) and c.1211_1212insA (p.N404Kfs) mutations of ATP7B
clinical manifestation, physical examination	mild jaundice—fatigue neuropsychiatric symptoms hypomimic face monotonic and slow speech increased muscle tonus postural tremor of extremities broad based and ataxic gait	no history of hepatic, neurological or psychiatric symptoms
slitlamp (KF-ring)	bilateral KF-ring	negative for KF-ring
abdominal US exam	areas of increased nonhomogeneous echogenicity	normal
liver biopsy	scars and regenerative nodules with inflammatory infiltrates liver cirrhosis	n.a.
MRI	hyperintensive areas on T2- weighted images in the BG cortical and subcortical brain atrophy	normal
Twin pair 2		
genetic testing	homozygous missense mutation c.3207C>A (p.H1069Q)	homozygous missense mutation c.3207C>A (p.H1069Q) of ATP7B
clinical ma-nifestation, physical examination	mild dysarthria slight paresis of left upper arm slight ataxia, broad based gait	no history of hepatic, neurological or psychiatric symptoms
slitlamp (KF-ring)	bilateral KF-ring	less saturated bilateral KF-ring
abdominal US exam	hepatosplenomegaly within the left lobe, multiple hyperechogenic lesions	hepatosplenomegaly dilated portal vein (11.9 mm)
cMRI	increased signal in T2-weighted images of BG, thalamus, mesencephalon, pons and cerebral peduncle, distinct atrophy of cerebellum and features of brainstem atrophy	increased signal in T2-weighted images of the lenticular ganglia, thalamus, cerebral peduncle and pons
Publication 3: Kegley et al. [28]		
genetic testing	homozygous mutation (H1069Q) of ATP7B	homozygous mutation (H1069Q) of ATP7B
clinical manifestation, physical examination	generalized malaise fatigue and abdominal pain hepatic encephalopathy	no sign and symptoms
liver biopsy (10–35 lg/g dry weight)	micronodular cirrhosis with prominent ductular proliferation, cholestasis, mild steatosis and ongoing hepatocyte necrosis quantitative copper: 2241 lg/g dry weight in the explanted liver)	grade 2 inflammation with stage 1 to 2 fibrosis quantitative copper: 1916 lg/g dry weight
treatment	denied any medication	firstly denied any medication then copper chelating agent
liver transplantation	successful orthotopic liver transplantation	n.a.

Details of treatment are highlighted in Table 3.

## Data Availability

Data available on request due to restrictions (e.g., privacy or ethical). The data presented in this study are available on request from the corresponding author.

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
