# Peer review of "Different Response Behavior to Therapeutic Approaches in Homozygotic Wilson’s Disease Twins with Clinical Phenotypic Variability: Case Report and Literature Review"

_genes, 2022, doi:10.3390/genes13071217_

Round 1

Reviewer 1 Report

The case described in the article is certainly unique. There are only isolated descriptions of the course of hereditary diseases in pairs of monozygotic twins. The accumulation of information on the discordance of symptoms of monogenic diseases in people with the same genotype provides basic information for assessing the penetrance and expressiveness of monogenic diseases. I really liked the presentation of the biochemical and clinical phenotype of probands in the form of graphs.

I would recommend shortening the article for a more visual representation of the clinical case:

1)Some of the material from the introduction should be transferred to the discussion. In the introduction there is textual information about such clinical cases, and in the discussion the same information is more detailed in the form of a table, but without the actual discussion in the text

2)To present the historical information about siblings in a more structured way, in the form of a table, supplemented with text comments explaining the information.

Comments on the merits of the work:

1)Information on the method of molecular genetic tests of sisters is not provided;

2) There is no evidence of monozygosity of twins in the work. Has a STR or SNP analysis been carried out? Without proof of zygosity, this case should be considered as a common diverse course of the disease in women with the same mutations.

3) There is no data on whether the search for any de novo variants allowing to explain such a severe toxic effect of therapy on the kidneys was carried out for sibs with a severe clinical picture.

4) The second sibs cannot be fully called asymptomatic, since she definitely had biochemical symptoms, although milder.

5) The article does not provide convincing arguments about the different susceptibility of siblings to therapy. Both sisters responded well to therapy with high doses of D-penicillamine, but in one of the sisters it had a toxic effect.

Additionally, I want to note that in the discussion section, you can speculate that one of the differences between the sisters was the pregnancy of one of them at a fairly young age. During pregnancy, the concentrations of metals in the blood and the susceptibility of target organs to them are modify.

I believe that this unique case can be published after some refinement (reduction and addition of molecular genetic data).

Author Response

The case described in the article is certainly unique. There are only isolated descriptions of the course of hereditary diseases in pairs of monozygotic twins. The accumulation of information on the discordance of symptoms of monogenic diseases in people with the same genotype provides basic information for assessing the penetrance and expressiveness of monogenic diseases. I really liked the presentation of the biochemical and clinical phenotype of probands in the form of graphs.

I would recommend shortening the article for a more visual representation of the clinical case:

1)Some of the material from the introduction should be transferred to the discussion. In the introduction there is textual information about such clinical cases, and in the discussion the same information is more detailed in the form of a table, but without the actual discussion in the text

2)To present the historical information about siblings in a more structured way, in the form of a table, supplemented with text comments explaining the information.

Comments on the merits of the work:

1)Information on the method of molecular genetic tests of sisters is not provided;

2) There is no evidence of monozygosity of twins in the work. Has a STR or SNP analysis been carried out? Without proof of zygosity, this case should be considered as a common diverse course of the disease in women with the same mutations.

3) There is no data on whether the search for any de novo variants allowing to explain such a severe toxic effect of therapy on the kidneys was carried out for sibs with a severe clinical picture.

4) The second sibs cannot be fully called asymptomatic, since she definitely had biochemical symptoms, although milder.

5) The article does not provide convincing arguments about the different susceptibility of siblings to therapy. Both sisters responded well to therapy with high doses of D-penicillamine, but in one of the sisters it had a toxic effect.

Additionally, I want to note that in the discussion section, you can speculate that one of the differences between the sisters was the pregnancy of one of them at a fairly young age. During pregnancy, the concentrations of metals in the blood and the susceptibility of target organs to them are modify.

I believe that this unique case can be published after some refinement (reduction and addition of molecular genetic data).

Reviewer 1 is right:

We now present details on previously reported homozygous WD-twins with different phenotyp only in the discussion.

The history of the siblings before visit to our institution is now summarized in Table 1. Table 2 is slightly shortened.

We rely on the written report of an external institution (Institut für Medizinische Genetik Köln) which explicitly mentions that both monozygotoc twins suffer from the identical mutations of WD.

This is now explicitly mentioned in the text.

Indeed, we did not find information on nephrotic syndrome in WD-twins.  

Reviewer 1 is right: we now avoid the term “completely asymptomatic” since that is not true.

Also in this aspect reviewer 1 is right: we have emphasized that both sisters responded to DPA.

The sister being treated with a 3fold higher dose had the toxic effect.

We are very thankful to reviewer 1 that he encouraged us to emphasize this aspect more clearly.

Reviewer 2 Report

Is there any anamnestic evidence in the twin sisters of a different environment or diet that can explain the different phenotypes? This could be referred to in the discussion.

Author Response

Is there any anamnestic evidence in the twin sisters of a different environment or diet that can explain the different phenotypes? This could be referred to in the discussion.

Different arguments are now summarized in the discussion and a special graph, which factors might have influenced the phenotype. From the time of their marriage on, the disease development is different in the two sisters. We therefore emphasize that change in life style, living place (especially the copper content in the water) and pregnancy may have caused the difference in the onset of WD manifestation.

Reviewer 3 Report

In the manuscript entitled “Different response behavior to therapeutic approaches in homozygotic Wilson’s disease twins with clinical phenotypic variability: case report and literature review”, the authors presented a case of WD-twins with phenotypic discordance and their different response behavior to WD-specific therapy. In the results, the authors also listed several findings of homozygotic twins in WD and discussed the pheno/genotype correlation.

The idea of this article is innovative and provides some reasons for genotype-phenotype discordance in the WD field. However, as a review, I think it has some defects in the following details:

1.     In the introduction, “More than 700 mutations have been described in ATP7B.” As far as I know, there are more than 1000 kinds of mutations in the HGMD database until 2021. This number is suggested to be renewed.

2.     As you mentioned in the case of sister 1, “After switching from DPA therapy to Trientine (Cuprior®), the protein loss, the general condition, and the neurological symptoms further improved”, It is suggested that you provide more details of the medicine, like dosage, drug reaction, etc.

3.     For the treatment of sister 1, I have a pretty negative attitude toward the DPA dosages (also opinions in discussion 4.2). Considering her unstable neurological symptoms, 1800mg DPA/day is highly possible to aggravate the patient's condition and cause other irreversible damage.

4.     For clinical evaluation of the patients, I recommend using the more well-known neurological assessment scale like UWDRS, semi-quantitative MRI scale, etc.

5.     In your discussion, you mentioned the effect of methylation metabolism and whether it was possible to measure methylation in both patients?

6.     In the manuscript, the authors used many long sentences, which were obscure. The language should be improved.

Author Response

In the manuscript entitled “Different response behavior to therapeutic approaches in homozygotic Wilson’s disease twins with clinical phenotypic variability: case report and literature review”, the authors presented a case of WD-twins with phenotypic discordance and their different response behavior to WD-specific therapy. In the results, the authors also listed several findings of homozygotic twins in WD and discussed the pheno/genotype correlation.

The idea of this article is innovative and provides some reasons for genotype-phenotype discordance in the WD field. However, as a review, I think it has some defects in the following details:

1.     In the introduction, “More than 700 mutations have been described in ATP7B.” As far as I know, there are more than 1000 kinds of mutations in the c. This number is suggested to be renewed.

2.     As you mentioned in the case of sister 1, “After switching from DPA therapy to Trientine (Cuprior®), the protein loss, the general condition, and the neurological symptoms further improved”, It is suggested that you provide more details of the medicine, like dosage, drug reaction, etc.

3.     For the treatment of sister 1, I have a pretty negative attitude toward the DPA dosages (also opinions in discussion 4.2). Considering her unstable neurological symptoms, 1800mg DPA/day is highly possible to aggravate the patient's condition and cause other irreversible damage.

4.     For clinical evaluation of the patients, I recommend using the more well-known neurological assessment scale like UWDRS, semi-quantitative MRI scale, etc.

5.     In your discussion, you mentioned the effect of methylation metabolism and whether it was possible to measure methylation in both patients?

6.     In the manuscript, the authors used many long sentences, which were obscure. The language should be improved.

Reviewer 3 is right: we checked the database and have changed the number of mutations to more than 1000.

Details of the switch of the drug are now presented.

This point is well taken. Sister 1 responded well until she developed the nephrotic syndrome. We now shortly commend that high doses of DPA may worsen symptoms in WD. But we do not think that did happen in the present case.

One of us was coauthor when the UWDRS was introduced. In clinical practice, UWDRS is time consuming, and factor analysis of the UWDRS shows that some items our not independent on each other or are redundant, in contrast to our score. Our score can be completed within a minute after a neurological examination. In clinical practice, we exclusively use our score.

We do not measure methylation in our WD-patients.

We tried to improve language and improve readability.   

Reviewer 4 Report

In this work, the authors investigated the potential genotype/phenotype correlation in genetically identical WD-twins showing phenotypic discordance and different response behaviour to WD-specific therapy.

The topic and the results are interesting; however, my major concern regards the structure of the paper: is this a cases report or a review? It seems to me a case report and, although the authors can leave the literature revision, they should follow the structure of a case report, if appropriate.  

Minor points:

1. Abstract section: the authors should report the extension of DPA

2.  In the legend of Table 1, the authors should report the extension of OCT

3. Table 2 should be better designed and structured in the same way in all studies reported

4. Check for English errors or general typos: i.e geno/phenotyp in the conclusions section.

Author Response

In this work, the authors investigated the potential genotype/phenotype correlation in genetically identical WD-twins showing phenotypic discordance and different response behaviour to WD-specific therapy.

The topic and the results are interesting; however, my major concern regards the structure of the paper: is this a cases report or a review? It seems to me a case report and, although the authors can leave the literature revision, they should follow the structure of a case report, if appropriate. 

Minor points:

1. Abstract section: the authors should report the extension of DPA

2.  In the legend of Table 1, the authors should report the extension of OCT

3. Table 2 should be better designed and structured in the same way in all studies reported

4. Check for English errors or general typos: i.e geno/phenotyp in the conclusions section.

We now make the structure of the case report more clearly by addition of a further table and shifting of the old Table 2 to the discussion.

Extension of DPA is presented.

Extension of OCT is presented in the text and table.

Old Table 2 has become Table 3 now, is shifted to the discussion and has been reduced to 50% of its previous size.

Several typos and errors have been corrected.

Round 2

Reviewer 1 Report

The article looks very good. But the evidence of monozygosity of twins is still lacking. This is a very simple analysis - it is enough to genotype both sisters with any identity identification kit, for example AmpFlSTR Identifiler and show that the markers profiles are the same. It is incorrect to draw a conclusion about the monozygosity of twins only on the basis of identical mutations of one gene.

Author Response

The article looks very good. But the evidence of monozygosity of twins is still lacking. This is a very simple analysis - it is enough to genotype both sisters with any identity identification kit, for example AmpFlSTR Identifiler and show that the markers profiles are the same. It is incorrect to draw a conclusion about the monozygosity of twins only on the basis of identical mutations of one gene.

The mutations were performed in an external institute. The report of this institute explicitly mentions the monozygosity of the twins. We had no reason not to rely on this report.

If a scan of this report should be added as additional material we have to ask whether we are allowed to do this. This may take some days.  

Reviewer 3 Report

In this revised version, the author changed several language issues and clarified some contents more clearly. At the same time, the author gave good answers to reviewers’ questions. I agree to accept this manuscript.

Author Response

Thank you so much for your valuable comments.